# DrivingGen: A Comprehensive Benchmark for Generative Video World Models in Autonomous Driving

**Yang Zhou**[1][*]   **Hao Shao**[2][*]   **Letian Wang**[1]

**Zhuofan Zong**[2]   **Hongsheng Li**[2]   **Steven L. Waslander**[1]

[1] University of Toronto    [2] CUHK MMLab

Project Website: https://drivinggen-bench.github.io/

## Abstract

Video generation models, as one form of world models, have emerged as one of the most exciting frontiers in AI, promising agents the ability to imagine the future by modeling the temporal evolution of complex scenes. In autonomous driving, this vision gives rise to driving world models: generative simulators that imagine ego and agent futures, enabling scalable simulation, safe testing of corner cases, and rich synthetic data generation. Yet, despite fast-growing research activity, the field lacks a rigorous benchmark to measure progress and guide priorities. Existing evaluations remain limited: generic video metrics overlook safety-critical imaging factors; trajectory plausibility is rarely quantified; temporal and agent-level consistency is neglected; and controllability with respect to ego conditioning is ignored. Moreover, current datasets fail to cover the diversity of conditions required for real-world deployment. To address these gaps, we present DrivingGen, the first comprehensive benchmark for generative driving world models. DrivingGen combines a diverse evaluation dataset curated from both driving datasets and internet-scale video sources, spanning varied weather, time of day, geographic regions, and complex maneuvers, with a suite of new metrics that jointly assess visual realism, trajectory plausibility, temporal coherence, and controllability. Benchmarking 14 state-of-the-art models reveals clear trade-offs: general models look better but break physics, while driving-specific ones capture motion realistically but lag in visual quality. DrivingGen offers a unified evaluation framework to foster reliable, controllable, and deployable driving world models, enabling scalable simulation, planning, and data-driven decision-making.

## 1 Introduction

Driven by scalable learning techniques, generative video models have made remarkable progress in recent years, enabling the synthesis of high-fidelity videos across diverse scenes and motions. These models suggest a promising path toward "world models" – predictive simulators capable of imagining the future, which can support planning, simulation, and decision-making in complex, dynamic environments. Inspired by this vision, there has been an accelerating surge in developing driving world models: generative models specialized for predicting future driving scenarios. Given an initial scene and optional conditions (*e.g.*, text prompts, driving actions), a driving world model predicts both the ego-vehicle's future movements and the evolution of surrounding agents' trajectories. Such models enable closed-loop simulation and synthetic data generation, reducing reliance on real-world data and offering a promising means to explore out-of-distribution scenarios safely (Gao et al., 2024; Hassan et al., 2024; Mousakhan et al., 2025; Li et al., 2025d; Wang et al., 2025; Zhou et al., 2025). Driving world models are also tightly coupled with end-to-end autonomous driving systems, where

---

[*]Equal contribution.

errors in predicted future scenes and trajectories can directly lead to unsafe decisions (Shao et al., 2023a; 2024a; 2023b; Wang et al., 2023).

| Method / Benchmark | Evaluation Metrics | | | | |
|---|---|---|---|---|---|
| | Distribution | Quality | Temporal Consistency | Alignment | Downstream Task |
| VBench (Huang et al., 2023) | ✗ | ✔ | ✔ | | ✗ |
| WorldModelBench (Li et al., 2025a) | ✗ | ✔ | ✔ | Instruction | ✗ |
| WorldScore (Duan et al., 2025) | ✗ | ✔ | ✔ | Traj. | ✗ |
| Vista (Gao et al., 2024) | Visual | Human eval | ✗ | Traj. | ✗ |
| GEM (Hassan et al., 2024) | Visual | Human eval | Human eval, Agent | Traj. | ✗ |
| Doe-1 (Zheng et al., 2024c) | Visual | ✗ | ✗ | ✗ | VQA, Planning |
| Drivingdojo (Wang et al., 2024e) | Visual | ✗ | ✗ | Traj. | ✗ |
| Driverse (Li et al., 2025d) | Visual | ✗ | ✗ | Traj. | ✗ |
| UniFuture (Liang et al., 2025) | Visual | ✗ | ✗ | ✗ | Perception |
| VaViM (Bartoccioni et al., 2025) | Visual | ✗ | ✗ | ✗ | Segmentation |
| GAIA-2 (Russell et al., 2025) | Visual | ✗ | Visual, Agent | ✗ | ✗ |
| ReSim (Yang et al., 2025) | Visual | ✗ | ✗ | Traj. | Planning |
| ACT-Bench (Arai et al., 2024) | ✗ | ✗ | ✗ | Instruction, Traj. | ✗ |
| DrivingGen (Ours) | Visual, Traj. | Visual, Traj. | Visual, Agent, Traj. | Traj. | ✗ |

Table 1: Comparison of existing video benchmarks, driving world models, and driving video benchmarks. "✗" indicates the missing metrics, and "✔" signifies that the evaluation is comprehensive. "Visual", "Agent" and "Traj." represent evaluation of images or videos, surrounding agents and vehicles' trajectories, respectively.

While a vibrant exploration of a wide range of approaches for driving world models is underway, a well-designed benchmark – which not only measures progress but also guides research priorities and shapes the trajectory of the entire field – has not yet emerged. Current evaluations fail to fully capture the unique requirements of the driving domain, and are limited in several ways. 1) *Visual Fidelity* First, most benchmarks rely on distribution-level metrics such as Fréchet Video Distance (FVD) to assess video realism, and some adopt human-preference-aligned models (e.g., vision-language models) to score visual quality or semantic consistency. However, driving imposes unique constraints on imaging: sensor artifacts, glare, or other corruptions can have critical safety implications that general video metrics fail to capture. 2) *Trajectory Plausibility* Second, the ego-motion trajectories underlying the generated videos are crucial. High-quality video generation in driving must produce trajectories that are natural, dynamically feasible, interaction-aware, and safe—properties that go beyond mere visual realism. 3) *Temporal and Agent-Level Consistency* Third, temporal consistency is crucial for driving, where surrounding objects directly impact safety and decision-making. Prior benchmarks often focus on scene-level consistency but neglect agent-level consistency, such as abrupt appearance changes or abnormal disappearances of agents—imperfections that can severely compromise the realism and reliability of driving simulations. 4) *Motion Controllability* Finally, for ego-conditioned video generation, it is critical to assess whether the generated motion faithfully follows the conditioning trajectory. This aspect of controllability is largely overlooked in existing benchmarks, yet it is essential for safe planning and reliable closed-loop driving, where misalignment can lead to catastrophic consequences.

Another major limitation in existing benchmarks for driving world models is the lack of diversity along crucial dimensions essential for real-world deployment. 1) First, *Weather and Time of Day* coverage is heavily skewed: datasets like nuScenes (Caesar et al., 2020) are dominated by clear-weather, daytime driving, leaving rare but safety-critical conditions (night, snow, fog) underrepresented. 2) Second, *Geographic Coverage* is limited, often confined to a few cities or countries, which restricts evaluation across varied scene appearance and with local traffic rules. 3) Third, *Driving Maneuvers and Interactions* rarely capture the full diversity of agent behaviors and complex multi-agent dynamics, such as pedestrians waiting at crosswalks, aggressive driver cut-ins, or dense traffic scenarios (Wang et al., 2021). This lack of diversity makes it difficult to assess whether generative models can handle the wide range of scenarios encountered in real-world driving, undermining their reliability and safety for large-scale deployment.

To address the above gaps, this work proposes DrivingGen, a comprehensive benchmark for generative world models in the driving domain with a diverse data distribution and novel evaluation metrics. DrivingGen evaluates models from both a visual perspective (the realism and overall quality of generated videos) and a robotics perspective (the physical plausibility, consistency and accuracy of generated trajectories). Our benchmark makes the following key contributions:

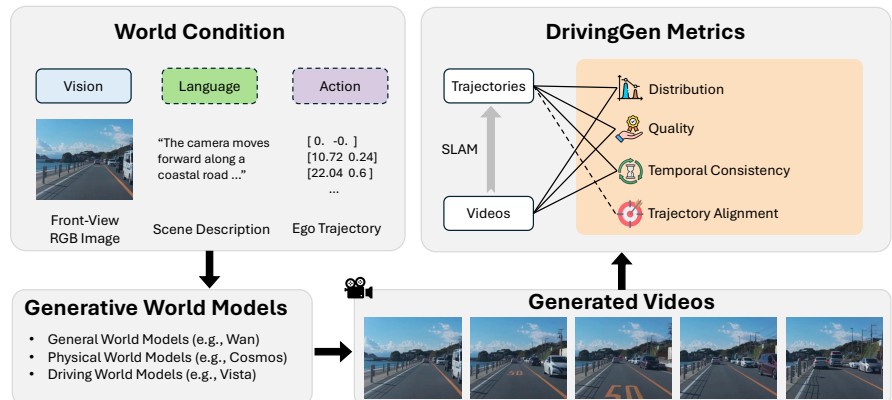

Figure 1: Overview of our DrivingGen benchmark. Video models take vision, and optional language/action as inputs to generate videos. The generated videos are then passed into our evaluation suite. Four comprehensive and novel sets of metrics for both videos and trajectories (distribution, quality, temporal consistency, and trajectory alignment) are introduced to evaluate world models.

**Diverse Driving Dataset**. We present a new evaluation dataset that captures diverse driving conditions and behaviors. Unlike prior datasets biased toward sunny, daytime urban scenes, ours includes varied weather (rain, snow, fog, floods, sandstorms), times of day (dawn, day, night), global regions (North America, Europe, Asia, Africa, etc.), and complex scenarios (dense traffic, sudden cut-ins, pedestrian crossings). This diversity enables more robust and unbiased evaluation of generative models under realistic driving distributions. Besides, considering that inference for video generation is generally time-consuming, we carefully limit the number of samples to 400 to ensure efficient testing and iteration, achieving a balance between efficiency and meaningful evaluation.

**Driving-Specific Evaluation Metrics**. We introduce a novel suite of multifaceted metrics specifically designed for driving scenarios. These include distribution-level measures for both video and trajectory outputs, quality metrics that account for human perceptual quality, driving-specific imaging factors (such as illumination flicker, motion blur, etc.), temporal consistency checking at both the scene level and individual agent level (*e.g.*, appearance discrepancy or unnatural disappearances in videos), and trajectory realism metrics that evaluate kinematic feasibility and alignment to intended paths (*e.g.*, smoothness, physical plausibility, and accuracy in following a given route). Together, these metrics provide a comprehensive 4-dimensional evaluation along distribution realism, visual quality, temporal coherence, and control/trajectory fidelity – covering aspects that generic metrics or single-number scores fail to capture.

**Extensive Benchmarking and Insights**. We benchmark 14 generative world models on DrivingGen spanning three categories – general video world models, physics-based world models, and driving-specialized world models. This evaluation, the first of its kind in the driving domain, reveals important insights and open challenges. For example, we find that certain general world models produce visually appealing traffic scenes yet break physical consistency in vehicle motion, and some driving-specific models excel in trajectory accuracy but lag in image fidelity. By analyzing performance across our metrics, we reveal the strengths and failure modes of each approach, offering insights for future research. All components of DrivingGen—dataset and evaluation code—are publicly released to support reproducible research and advance realistic driving simulation.

## 2 RELATED WORKS

In this work, we focus on two primary research areas: generative world models applied to autonomous driving and benchmarks for evaluating these models. Due to space constraints, we provide a comprehensive review of the relevant literature, including recent advancements in general video generation and specific driving-world evaluations, in Appendix A.

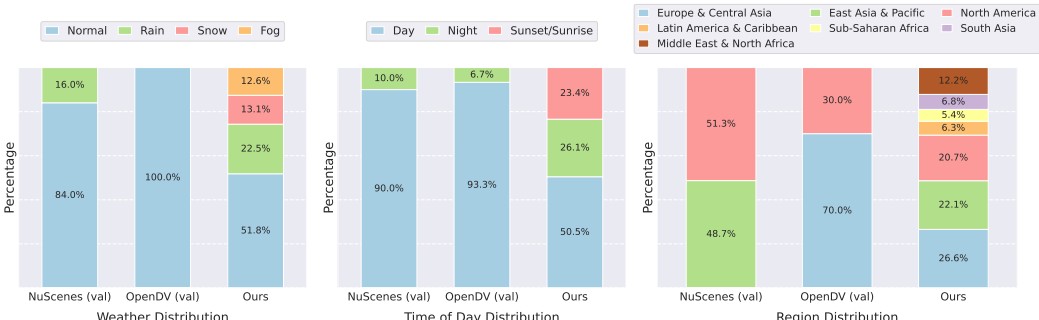

(a) Weather, time of day, and region distribution between existing datasets and ours.

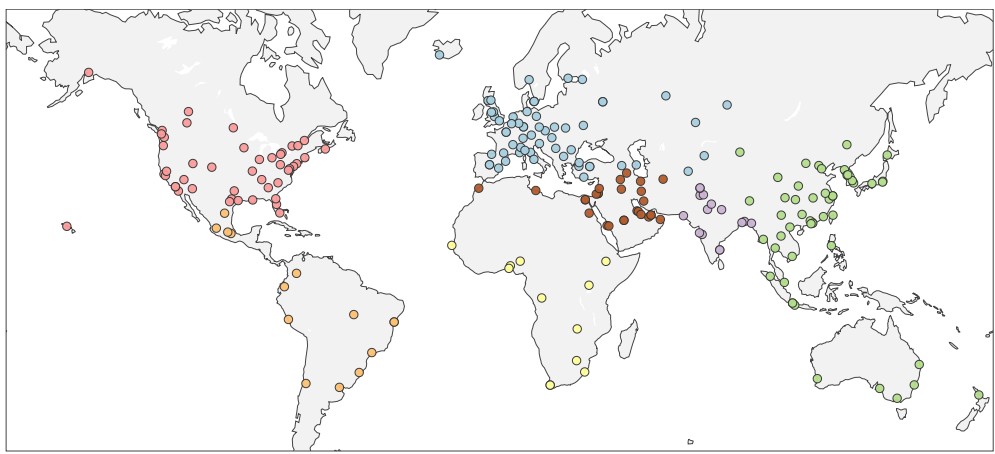

(b) Specific driving locations inside each region of our dataset.

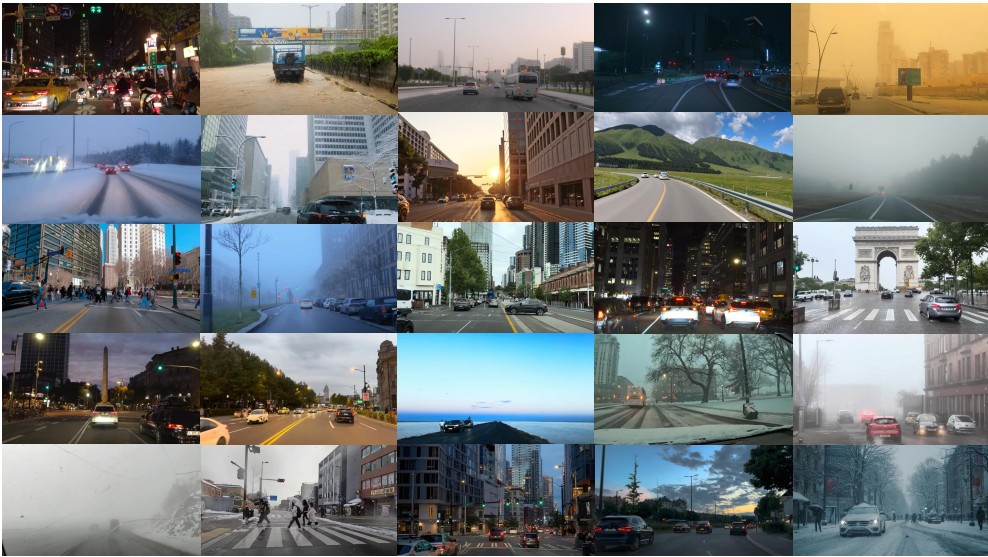

(c) Representative examples in our benchmark, which covers diverse scenarios such as dense city traffic at night, unusual weather (*e.g.*, fog, flood, sandstorm), and complex interactions (*e.g.*, waiting for pedestrians, agents cutting in).

Figure 2: Dataset distribution and gallery in our benchmark (top to bottom).

## 3 DRIVINGGEN BENCHMARK

The goal of DrivingGen is to establish a comprehensive benchmark to evaluate generative world models under driving-specific constraints and criteria. To achieve this, the proposed benchmark includes several key components: 1) a carefully collected dataset that is diverse in weather, time of day, regions (and their driving styles), and driving maneuvers to support reasonable evaluation; 2) multifaceted metrics that not only evaluate the video quality from a general visual perspective (*e.g.*, appearance), but also from a driving and robotics perspective (*e.g.*, the physical feasibility of trajectories). To showcase the distinguishing capability of DrivingGen, we evaluate general world models, physics-based models, and driving-specific models. An overview is given in Fig. 1, with dataset details in Sec. 3.1 and metrics in Sec. 3.2.

### 3.1 BENCHMARK DATASET

Generative video models, as a form of world models, offer a promising way to anticipate future driving scenarios, simulate rare or safety-critical events, and ultimately support planning and decision-making. However, real-world driving unfolds under highly variable conditions, encompassing different weather, lighting, regions, and complex maneuvers. Therefore, evaluating generative models across diverse scenarios is crucial to ensure their robustness and reliability. To this end, the majority of existing works (Gao et al., 2024; Hassan et al., 2024; Liang et al., 2025; Wang et al., 2024e; Bartoccioni et al., 2025; Wang et al., 2024b) in driving world models mainly utilize nuScenes (Caesar et al., 2020) and OpenDV (Zheng et al., 2024b) datasets for evaluation. However, the diversity of weather, region, time of day, and driving maneuvers in these datasets is limited and highly biases the data distribution. For example, as shown in Fig. 2a, over 80% of the nuScenes validation data and 90% of the OpenDV validation data are collected during normal sunny daytime conditions. Additionally, the data are collected from a limited number of vehicles and locations, which further limits the comprehensiveness. Based on this observation, we curated a significantly more diverse dataset. An overview of our dataset is presented in Fig. 2a and Fig. 2b.

**Dataset Construction.** We organize our dataset into two complementary tracks, offering distinct perspectives for evaluating driving videos.

- *Open-Domain Track* is designed to evaluate models' generalization to open-domain, diverse, unseen driving scenarios. We construct this track using Internet-sourced data spanning multiple cities and regions worldwide, ensuring broad coverage beyond the training distribution.

- *Ego-Conditioned Track* complements the open-domain track. While the open-domain setting evaluates generalization to diverse unseen scenarios, it does not verify whether the generated trajectories follow a specified conditioning trajectory—a property that is critical for robotics and self-driving applications. The ego-conditioned track therefore focuses on trajectory controllability, measuring how well the trajectories derived from generated videos align with the given ego-trajectory instructions. The ego trajectory is optional for model input and only provided in this track. To construct it, we aggregate data from five open-source driving datasets: Zod (Alibeigi et al., 2023) (Europe), DrivingDojo (Wang et al., 2024e) (China), COVLA (Arai et al., 2025) (Japan), nuPlan (Karnchanachari et al., 2024) (US), and WOMD (Sun et al., 2020) (US).

Each data sample in the dataset consists of three components: a front-view RGB image (vision), a scene description (language), and an optional ego trajectory (action). For each scene, we employ Qwen (Bai et al., 2025) to capture descriptions of the future dynamics and camera movements within the scene. Given the time-consuming nature of video generation, we limit the number of samples for efficient testing and iteration, while ensuring quality and diversity. The dataset includes 400 samples—200 per track—striking a balance between efficiency and meaningful evaluation.

**Balanced Data Dsitribution** The overall distribution of our dataset, along with a gallery of representative video examples, is shown in Fig. 2. To ensure meaningful evaluation, we explicitly control diversity across several dimensions:

- *Weather and Time of Day.* Existing benchmarks are often dominated by, if not fully composed of, normal weather and daytime conditions. In contrast, our benchmark aims for a more balanced distribution. For the open-domain track, we limit normal weather and daytime clips to below 60% and increase the proportion of other conditions, such as snow (13.1%), fog (12.6%),

| Distribution | Quality | Temporal Consistency | Trajectory Alignment |
|---|---|---|---|
| 1. Fréchet Video Distance (FVD)
2. Fréchet Trajectory Distance (FTD) | 1. Subjective Image Quality
2. Objective Image Quality
3. Trajectory Quality | 1. Video Consistency
2. Agent Consistency
3. Agent Disappearance Consistency
4. Trajectory Consistency | 1. Average Displacement Error (ADE)
2. Dynamic Time Warping (DTW) |

Table 2: Overview of metrics utilized in DrivingGen. Definition and details are in Sec. 3.2.

and night/sunset/sunrise driving (50%), to ensure a more comprehensive evaluation. Extreme events, including sandstorms, floods, and heavy snowfall at night, are also included. A similar strategy is applied to the ego-conditioned track, where normal weather/daytime clips make up 60% of the data, while the remainder covers diverse conditions to support trajectory controllability evaluation across different scenarios.

- *Geographic Coverage.* Prior benchmarks are often limited to a small number of cities or countries, restricting the diversity of driving scenarios. For the open-domain track, we collect data from a wide range of regions worldwide, including North America (20.7%), East Asia & Pacific (22.1%), Europe & Central Asia (26.6%), the Middle East & North Africa (12.1%), Latin America & Caribbean (6.3%), South Asia (6.8%) and South-Saharan Africa (5.4%), to ensure broad geographic coverage. For the ego-conditioned track, data are drawn from existing datasets covering North America, Asia and Europe, providing diverse driving scenarios to evaluate ego-trajectory alignment and controllability.

- *Driving Maneuvers and Interactions.* Capturing diverse driving behaviors and multi-agent interactions is critical for evaluating generative world models. For the open-domain track, scenarios include complex interactions such as waiting pedestrians at crosswalks, other agents cutting in, and dense traffic, testing the model's understanding of the driving world. For the ego-conditioned track, scenarios are similarly diverse, emphasizing multi-agent interactions and challenging conditions to evaluate controllability and alignment with ego-trajectory instructions.

## 3.2 BENCHMARK METRICS

For all video models, our DrivingGen metrics cover three key dimensions: distribution, quality, and temporal consistency, evaluated for both videos and trajectories. We extract trajectories using standard PnP method within a SIFT and RANSAC scheme (Lowe, 2004; Fischler & Bolles, 1981; Kneip et al., 2011) and UniDepthV2 (Piccinelli et al., 2025). We provide the details of our SLAM pipeline (Mur-Artal et al., 2015; Schönberger & Frahm, 2016; Teed & Deng, 2022; Qu et al., 2024), including guaranteeing that all videos reconstruct trajectories and a discussion to compare other benchmarks' trajectory reconstruction methods in Appendix B.2. For models conditioned on ego trajectories, we include a fourth dimension: trajectory alignment, measuring adherence to the input. Table 2 lists the metrics, grouped into four categories detailed below, each targeting a different aspect of video fidelity.

### 3.2.1 DISTRIBUTION

*How far is the generative distribution from the data distribution?* A common practice is to measure Fréchet Video Distance (FVD) (Unterthiner et al., 2019) on generated videos. However, our key insight is that video quality is not solely determined by visual realism—equally important, especially for self-driving and embodied agents, is the realism of the induced ego-motion. Focusing only on visual fidelity gives an incomplete picture. Therefore, we evaluate distributional closeness across both videos and trajectories, capturing complementary perspectives from visual perception and robotics.

For the video distribution, we utilize FVD to quantify the similarity between generated videos and real videos. Specifically, we follow the standardized computation protocol from the original StyleGAN-V (Skorokhodov et al., 2022). For the trajectory distribution, we introduce a novel metric, Fréchet Trajectory Distance (FTD), a distributional metric tailored for evaluating driving trajectories. The key requirement is a trajectory encoder that maps trajectories into a latent space suitable for measuring distributional distance. To this end, we draw from the motion prediction domain—where models themselves are generative of future trajectories, and adopt the encoder of Motion Transformer (MTR) (Shi et al., 2023) as our encoding model. Details of FTD computation are provided in Appendix B.3.

### 3.2.2 QUALITY

*How good are the generated videos and trajectories?* To evaluate the fidelity of generated videos and trajectories in driving scenarios, we propose a comprehensive quality suite covering three aspects: perceptual video quality, domain-specific video quality, and trajectory quality.

**Visual Quality**. A common practice in generative video evaluation is to assess general perceptual quality with automatic, reference-free estimators aligned with human judgments. Specifically, we adopt CLIP-IQA+ (Wang et al., 2022), which leverages CLIP's vision-language representations to predict perceptual quality scores consistent with human subjective assessments. While effective, such subjective perceptual quality does not always align with what matters for driving, which unfolds outdoors, involves multiple agents, and occurs under real-world constraints. To additionally consider driving-specific imaging quality, we further adopt the Modulation Mitigation Probability (MMP) metric from the IEEE Automotive P2020 standard (Group et al., 2018; 996, 2022). MMP targets Pulse-Width Modulation (PWM)-induced flicker that can disrupt perception and tracking, and reports the fraction of time windows where residual temporal luminance modulation falls below a small threshold. Implementation details are in Appendix B.4.

**Trajectory Quality**. While prior evaluations often rely on video-based scores, they typically neglect whether the underlying motions are physically and kinematically plausible. To reduce the gap, DrivingGen introduces a composite, reference-free metric to assess the kinematic plausibility and ride comfort. Three individual submetrics are proposed and aggregated into a single score: 1) a comfort score penalizes extremes of longitudinal jerk, lateral acceleration, and yaw-rate, yielding a score to reward smoother, more comfortable motion; 2) a motion score that discourages under-mobility, as some trajectories barely move and stay static due to the model's weak ability; 3) a curvature score summarizes how much the path turns, discouraging zig-zags and unrealistically sharp bends. Together, these submetrics directly target properties that affect controllability, planning, and perceived comfort. Calculation details appear in Appendix B.5.

### 3.2.3 TEMPORAL CONSISTENCY

*How temporally consistent is the generated world?* We assess the temporal consistency of both videos and trajectories. For videos, we evaluate scene-level consistency, agent-level consistency, and explicitly emphasize abnormal agent disappearance. For trajectories, we measure the consistency of speed and acceleration over time, independent of path shape and absolute mobility.

**Video Consistency**. Existing metrics directly calculate the consistency between consecutive frames (or each frame to the first) at a fixed rate. However, it is easily hackable by generating near-static videos. To measure temporal consistency while accounting for the actual motion in the scene, we first pass the generated videos through an off-the-shelf optical flow model (Wang et al., 2024d) to compute the median optical flow magnitude per frame. We then adaptively downsample: videos with lower motion are sampled more sparsely so that the per-step displacement becomes comparable to normal/high-speed driving. After this, the similarity of the DINOv3 (Siméoni et al., 2025) features between consecutive frames of the downsampled videos is reported as the video consistency score. Unlike fixed-stride metrics, our approach fairly measures temporal consistency across videos with varying motion speeds, preventing static or near-static videos from obtaining artificially high scores.

**Agent Appearance Consistency**. Measuring only scene-level features can overlook small temporal changes in individual agents, such as shifts in color, texture, or shape, while these agents are often the key focus for driving, as they would more directly impact driving behavior and safety. To measure the agent's temporal consistency, we therefore detect agents in the first frame, track them across the video, crop their bounding boxes, and compute consistency purely at the agent level. We use YOLOv10 (Wang et al., 2024a) as the detector and SAM2 (Ravi et al., 2024; Yang et al., 2024a) for tracking. We measure DINOv3 feature similarity across consecutive frames and to the first frame.

**Agent Abnormal Disappearance**. In addition to appearance stability, agents in driving scenes must persist in a physically plausible manner. Sudden, non-physical disappearances of surrounding agents are commonly observed in generated videos, which can compromise realism and safety. DrivingGen quantifies this by diagnosing whether an agent's disappearance is normal (*e.g.*, leaving the field of view or being occluded) or abnormal. We consider three key frames for each disappearing agent: the first and the last frames where the agent is visible, and the first frame after it vanishes. A vision large language model (VLM) (Bai et al., 2025; Shao et al., 2024b; Liu et al., 2024; Zong et al.,

2024b; Li et al., 2024; Qu et al., 2025b;a), Cosmos-Reason1 (NVIDIA et al., 2025), is prompted to judge disappearance based on visual and motion continuity, and the agent's local interactions with surrounding agents. We report the percentage of videos with no abnormal disappearances as the score. Implementation Details can be found in Appendix B.6.

**Trajectory Consistency**. Realistic driving exhibits predictable kinematics: speed varies slowly around a cruise level and acceleration does not oscillate. To reveal this property, we compute how stable a trajectory's velocity and acceleration are over time. The average of the two scores is taken as the overall trajectory consistency score. Trajectories that jitter, stop–go, or oscillate score low, while steady cruising with gradual changes scores high. Calculation details are provided in Appendix B.7.

### 3.2.4 TRAJECTORY ALIGNMENT

In addition to trajectory consistency, the alignment of the trajectories underlying the generated videos with the conditioning (ego) trajectory is also critical, especially for trajectory-grounded video generation. To assess this, we propose two complementary metrics.

**Average Displacement Error** (ADE). As a common practice, ADE measures the mean pointwise distance between the generated and input trajectories across the prediction horizon. It emphasizes local, step-by-step fidelity and is standard in motion prediction and planning.

**Dynamic Time Warping** (DTW). In addition to ADE, which compares trajectories at each time step, we introduce a complementary metric that captures the overall contour and shape of the trajectory. Specifically, DTW (Keogh & Pazzani, 2000) aligns predicted and reference trajectories via non-linear time warping and measures their path-shape discrepancy using Euclidean point-wise cost.

## 4 EXPERIMENTS

**Evaluation Setup.** We evaluate 14 competitive generative world models on DrivingGen, spanning three categories. 1) First, we include 7 general video world models, comprising two commercial closed-source models, Gen-3 (Runway, 2024.06) and Kling (Kuaishou, 2024.06), and five well-known open-source models: CogVideoX (Yang et al., 2024e), Wan (Wan et al., 2025), Hunyuan-Video (Kong et al., 2024), LTX-Video (HaCohen et al., 2024a), and SkyReels (Chen et al., 2025). 2) Second, we evaluate 2 physical world models that are developed specifically for the physical robotics domain, Cosmos-Predict1 (Agarwal et al., 2025) and Cosmos-Predict2 (Cosmos, 2025). 3) Third, we assess 5 driving-specific world models: Vista (Gao et al., 2024), DrivingDojo (Wang et al., 2024e), GEM (Hassan et al., 2024), VaViM (Bartoccioni et al., 2025), and UniFuture (Liang et al., 2025). All models are evaluated on a prediction horizon of 100 frames. We report the time and resource cost for our DrivingGen benchmark in Appendix B.8.

### 4.1 OBSERVATIONS AND CHALLENGES

Table 3 presents the results. We provide the full table of metrics in a transparent way to evaluate the models comprehensively, and the average rank serves as a quick summary but not a definitive score. We also show that our results align well with human judgement, by calculating the Spearman's correlation coefficient (see details in Appendix B.9.) In the following, we will discuss key findings from our results.

**Closed-source models lead in visual quality and overall ranking.** Across both tracks, closed-source models consistently occupy the top positions, achieving strong perceptual scores and maintaining stable agent behavior. They rarely exhibit abnormal object disappearance and generally preserve scene coherence over time, demonstrating robust overall world generation capabilities.

**Top open-source general world models are competitive on specific metrics.** Several open-source models approach or match the closed-source leaders on individual dimensions. For example, CogVideoX and Wan achieve strong video distributional realism (low FVD) across both tracks, suggesting that open-source models can excel in targeted aspects even if they do not lead overall.

**No single model excels in both visual realism and trajectory fidelity.** We observe distinct "personas": some models achieve high visual quality but only moderate trajectory adherence and per-agent consistency, while driving-specialized models accurately follow commanded paths with phys-

| Open-Domain Track | | Distribution | | Quality | | | Temporal Consistency | | | | Avg. Rank |
|---|---|---|---|---|---|---|---|---|---|---|---|
| Models | Size | FVD | FTD | Subjective Quality | Objective Quality | Trajectory Quality | Video Consist | Agent Consist | Agent Missing | Trajectory Consist | |
| Kling 2.1* | - | 693.4 | **26.73** | **0.5538** | 0.8018 | 0.6438 | 0.8945 | 0.7981 | 0.9442 | **0.5377** | 1 |
| Gen-3 Alpha Turbo* | - | 801.0 | 93.50 | 0.5456 | 0.8378 | **0.6535** | 0.8900 | **0.8170** | 0.9495 | 0.4788 | 2 |
| LTX-Video | 13B | 648.2 | 31.29 | 0.5215 | 0.8288 | 0.5562 | 0.8851 | 0.7449 | 0.8977 | 0.4517 | 3 |
| Wan2.2-I2V | 14B | 609.0 | 63.86 | 0.5348 | 0.6396 | 0.5983 | 0.8883 | 0.7514 | 0.9128 | 0.4639 | 4 |
| HunyuanVideo-I2V | 13B | 957.5 | 30.95 | 0.4921 | 0.7207 | 0.4613 | 0.8821 | 0.8008 | 0.9306 | 0.4157 | 5 |
| SkyReels-V2-I2V | 14B | 876.0 | 52.93 | 0.5134 | 0.7432 | 0.4799 | 0.8776 | 0.7329 | 0.9078 | 0.4326 | 7 |
| CogVideoX | 5B | 621.2 | 236.7 | 0.4932 | 0.6802 | 0.3856 | 0.8211 | 0.7581 | 0.7661 | 0.2949 | 12 |
| Cosmos-Predict2 | 14B | 524.1 | 83.20 | 0.4931 | 0.7568 | 0.5990 | 0.8597 | 0.5912 | 0.8657 | 0.3997 | 8 |
| Cosmos-Predict1 | 14B | 821.1 | 81.22 | 0.5083 | 0.7207 | 0.2723 | 0.8429 | 0.6789 | 0.8796 | 0.2631 | 13 |
| Vista | 2.5B | 675.7 | 54.66 | 0.4340 | 0.8468 | 0.6030 | 0.8565 | 0.6357 | 0.8211 | 0.4040 | 6 |
| VaViM | 1.2B | 1446.6 | 449.2 | 0.4691 | 0.8468 | 0.3118 | **0.9159** | 0.7721 | **0.9752** | 0.0914 | 9 |
| UniFuture | 3.0B | 774.3 | 50.66 | 0.4206 | **0.9054** | 0.4507 | 0.8799 | 0.5373 | 0.8310 | 0.3858 | 10 |
| GEM | 2.1B | 770.1 | 147.1 | 0.5168 | 0.8423 | 0.5398 | 0.8176 | 0.6099 | 0.7788 | 0.3392 | 11 |
| Drivingdojo | 2.3B | 810.4 | 126.74 | 0.4202 | 0.8333 | 0.4511 | 0.8480 | 0.6256 | 0.8303 | 0.2739 | 14 |

| Ego-Conditioned Track | | Distribution | | Quality | | | Temporal Consistency | | | | Trajectory Alignment | | Avg. Rank |
|---|---|---|---|---|---|---|---|---|---|---|---|---|---|
| Models | Size | FVD | FTD | Subjective Quality | Objective Quality | Trajectory Quality | Video Consist | Agent Consist | Agent Missing | Trajectory Consist | ADE | DTW | |
| Kling 2.1* | - | 320.5 | 23.74 | 0.5468 | 0.7838 | **0.6860** | 0.8929 | 0.8186 | 0.9712 | **0.5430** | 29.97 | 2310 | 1 |
| Gen-3 Alpha Turbo* | - | 555.9 | 24.72 | **0.5740** | 0.8604 | 0.6770 | 0.8747 | 0.7986 | 0.9466 | 0.4800 | 33.39 | 2749 | 3 |
| Wan2.2-I2V | 14B | **194.4** | 29.56 | 0.5084 | 0.6982 | 0.6419 | 0.8821 | 0.7561 | 0.9034 | 0.4849 | 27.39 | 1901 | 2 |
| LTX-Video | 13B | 378.1 | 61.09 | 0.4895 | 0.8604 | 0.5464 | 0.8705 | 0.7708 | 0.9020 | 0.4442 | 32.12 | 2505 | 6 |
| HunyuanVideo-I2V | 13B | 532.9 | **21.18** | 0.4741 | 0.6847 | 0.5542 | 0.8792 | 0.8240 | 0.9415 | 0.4771 | 33.80 | 2794 | 7 |
| CogVideoX | 5B | 307.1 | 166.6 | 0.4884 | 0.6937 | 0.4252 | 0.8167 | 0.7541 | 0.8981 | 0.3783 | 32.67 | 2413 | 10 |
| SkyReels-V2-I2V | 14B | 428.2 | 57.02 | 0.4764 | 0.6622 | 0.5028 | 0.8661 | 0.7208 | 0.875 | 0.4322 | 31.54 | 2594 | 11 |
| Cosmos-Predict2 | 14B | 260.5 | 56.26 | 0.4756 | 0.8198 | 0.6424 | 0.8428 | 0.6707 | 0.8986 | 0.4108 | 22.38 | 1490 | 4 |
| Cosmos-Predict1 | 14B | 345.2 | 34.96 | 0.4783 | 0.7505 | 0.3761 | 0.8229 | 0.7423 | 0.7961 | 0.3343 | 34.47 | 3084 | 13 |
| Vista | 2.5B | 392.8 | 27.33 | 0.4146 | 0.8198 | 0.6047 | 0.8741 | 0.6417 | 0.8676 | 0.4366 | **19.70** | **1216** | 5 |
| UniFuture | 3.0B | 654.6 | 37.17 | 0.4006 | **0.9685** | 0.5353 | 0.8759 | 0.5525 | 0.8759 | 0.4165 | 20.21 | 1352 | 8 |
| VaViM | 1.2B | 1222 | 103.6 | 0.4910 | 0.8694 | 0.1936 | **0.9428** | **0.8290** | **0.9725** | 0.0984 | 41.92 | 3863 | 9 |
| Drivingdojo | 2.3B | 586.5 | 35.73 | 0.4264 | 0.8198 | 0.4131 | 0.8419 | 0.6940 | 0.8439 | 0.2776 | 25.50 | 2142 | 12 |
| GEM | 2.1B | 579.9 | 97.70 | 0.4484 | 0.8018 | 0.5085 | 0.7886 | 0.6180 | 0.7463 | 0.2983 | 25.73 | 1982 | 14 |

Table 3: **Evaluation results of 14 generative world models on our benchmark.** Best results are in red region, second best are in orange region, and third best are in blue region. "*" indicates commercial closed-source models. Models fall into four categories: closed-source, open-source general video models, physical-world models, and driving-specific models.

ically plausible motion (low ADE/DTW) yet underperform in visual fidelity, exhibiting noticeable artifacts. Currently, no model successfully combines strong photorealism with precise, physically consistent motion, highlighting a key frontier for driving world generation.

**Trajectory alignment remains limited, revealing substantial gaps.** Under ego-trajectory conditioning, models exhibit significant ADE/DTW errors, indicating poor adherence to commanded paths. This can stem from two main factors: 1) artifacts in the generated videos (*e.g.*, texture repetition, blur, unstable geometry) that impair SLAM-based trajectory recovery, and 2) imperfect motion generation, where the model itself fails to follow the intended trajectory. These observations highlight that both video fidelity and trajectory modeling need further improvement.

**DrivingGen exposes failure modes hidden from prior single metric.** Existing benchmarks often rely solely on distribution-level metrics such as FVD to evaluate generated driving videos. While useful for assessing overall distribution similarity, good FVD/FTD alone does not necessarily imply plausible driving—videos can appear distribution-close yet exhibit stop–go jitter, identity drift, or non-physical disappearances. Similarly, high objective quality (e.g., low flicker) can coexist with poor subjective quality or unstable agent behavior. By jointly reporting distribution, perceptual quality, temporal consistency, and trajectory alignment, DrivingGen exposes these hidden failure modes and highlights precisely where each model falls short.

## 5 CONCLUSION

This work introduces DrivingGen, a comprehensive benchmark designed to evaluate generative world models for autonomous driving. DrivingGen integrates a diverse dataset spanning varied weather, time of day, global regions, and complex driving maneuvers with a multifaceted metric

suite that jointly measures visual realism, trajectory plausibility, temporal coherence, and controllability. By benchmarking a broad spectrum of state-of-the-art models, DrivingGen reveals critical trade-offs among visual fidelity, physical consistency, and controllability, providing clear insights into the strengths and limitations of current approaches. The benchmark establishes a unified and reproducible framework that can guide the development of reliable and deployment-ready driving world models, fostering progress toward safe and scalable simulation, planning, and decision making in autonomous driving.

## 6   FUTURE WORK AND LIMITATIONS

As DrivingGen is the first comprehensive benchmark for generative world models in autonomous driving, several intriguing ideas can be explored further in follow-up work.

**Expanding More Meaningful Data.** Currently, we collect 400 data samples (from the web and aggregated from existing driving datasets) to balance efficiency and practicality, because generating and evaluating videos is resource-intensive. With this limited number, we may not fully cover the long tail of driving scenarios. In future expansions, scaling up the dataset is an exciting future direction. As generative models become faster and datasets become more readily available, scaling up to thousands of clips is feasible and will further improve long-tail coverage.

**Interactive and Closed-Loop Simulation.** Ensuring reliable closed-loop performance (*e.g.*, for safe planning) is crucial for Autonomous Driving, and DrivingGen is a step toward that by first benchmarking open-loop predictive quality and realism. In the current work, all considered generative video world models are designed for open-loop video generation and no standardized closed-loop world generation framework exists yet. Performing a fair, unified closed-loop benchmark is infeasible at this stage. An exciting future direction is to consider closed-loop evaluation for driving world models (*e.g.*, integrating generative models into an interactive simulator like CARLA or combining with closed-loop dataset simulation like Navsim).

**Downstream Tasks Metrics and Enriching data modality.** DrivingGen focuses on metrics that directly measure video realism, physical consistency, and controllability in the generated footage itself. One complementary direction is to incorporate metrics from downstream tasks in Autonomous Driving (*e.g.*, how well an autonomous driving stack performs using synthetic videos). However, it may require collecting synchronized multi-camera footage and Map knowledge for a fair and meaningful benchmark. Our current dataset is limited to a single front-view camera feed, which poses challenges for more structural driving generation. A possible future direction is expanding the benchmark to multi-view video and sensor data (LiDAR, HD Map, etc.) to construct a more structured driving world generation and novel metrics (*e.g.*, view consistency) can be proposed.

**Evaluation of Scene Controllability and State Transformation.** Evaluating controllability over scene content (*e.g.*, controlling other agents, road layout in the scene) would be highly useful for autonomous applications. We did not include such metrics in our benchmark because implementing a unified evaluation for different models with scene-level control faces challenges both in model support and dataset complexity. Due to these challenges, we believe it is a great topic for driving world generation which controls scene content and map layout and assessing whether state transformations of the world model are reasonable. One could imagine controlling the presence or behavior of a pedestrian or the configuration of lanes, and checking if the model can follow those constraints.

**Counterfactual Reasoning Evaluation.** In our current benchmark, we did not explicitly evaluate counterfactual reasoning. The main reason is that DrivingGen focuses on real driving videos. We are limited to evaluating the scenarios that actually happened. One novel future direction would be counterfactual reasoning evaluation. One can introduce hypothetical events or modifications (like an astronaut on a horse crossing the road, or a car jumping off the ground to overtake other agents, and other unrealistic edge cases) and propose new metrics to check whether the model follows this counterfactual generation.

**Overall Score.** We provide the full table of metrics transparently to evaluate the models, and the average rank serves as a quick summary but not a definitive score. Exploration of a composite, single-index score is an interesting topic, which requires normalized distribution and alignment metrics (*e.g.*, FVD and ADE).

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

# A  RELATED WORKS

## A.1  GENERATIVE WORLD MODELS AND THEIR APPLICATION IN DRIVING

Driven by advances in image generative modeling (Kingma & Welling, 2013; Goodfellow et al., 2014; Esser et al., 2021; Ho et al., 2020; Peebles & Xie, 2023; Zong et al., 2024a; He et al., 2025), the landscape of large-scale video models has evolved significantly, particularly in diffusion-based frameworks. Closed-source models (Brooks et al., 2024; Kuaishou, 2024.06; LumaLabs, 2024.06; Runway, 2024.06; PixVerse, 2023; Capcut, 2024; MiniMax, 2024.09; Tongyi, 2024.09; PikaLabs, 2024.10; Shao et al., 2024c), mainly developed by major technology companies, aim at high-quality, professional video generation with extensive resources invested. Sora (Brooks et al., 2024), introduced by OpenAI, marked a significant leap in Video Generation. Open-source models (Rombach et al., 2022; Ho & Salimans, 2022; HaCohen et al., 2024b; Kong et al., 2024; Wan et al., 2025; Yang et al., 2024e; Agarwal et al., 2025), typically based on stable diffusion (Rombach et al., 2022) and flow matching (Li et al., 2025b), are quickly expanding and making real contributions to video generation as well. Wan (Wan et al., 2025), an open-source model, is widely used for video generation and has achieved SOTA results on many benchmarks. Recent years have also seen remarkable progress in both multimodal understanding and generation models (Li et al., 2025c; Zhang et al., 2025).

Besides general video generation, driving-focused generative models use sensor data such as lidar point clouds (Zheng et al., 2024a; Yang et al., 2024d) or images (Gao et al., 2024; Hassan et al., 2024; Hu et al., 2023; Wang et al., 2024c;f; Yang et al., 2024b; Zhao et al., 2025). Since this work emphasizes video generation, we focus on image-based methods. Early approaches before Vista (Gao et al., 2024) rely on multi-view RGB inputs and high-definition maps or 3D boxes, limiting generalization to new datasets and open-domain videos. Vista-based methods (Hassan et al., 2024; Li et al., 2025d; Mousakhan et al., 2025) simplify inputs to a single front-view image with optional ego trajectories, improving scalability to YouTube videos and enabling broader open-domain evaluation.

## A.2  BENCHMARKS FOR EVALUATING GENERATIVE WORLD MODELS

The rapid progress of open- and closed-source video generation has driven the creation of many benchmarks (Huang et al., 2023; 2024; Bansal et al., 2024; Ning et al., 2023; Liao et al., 2024; Fan et al., 2024; Wang et al., 2024g), such as VBench, which evaluates models with multifaceted metrics based on human-collected prompts. Recently, evaluations have expanded to open, dynamic, and complex world-simulation scenarios (Yue et al., 2025; Duan et al., 2025; Li et al., 2025d; Qin et al., 2024; Kwon et al., 2025). WorldScore (Duan et al., 2025) measures generated videos using explicit camera trajectory layouts. However, a comprehensive driving-world benchmark is still lacking due to limited test sample diversity, heterogeneous input modalities, and the absence of driving-specific metrics. Recent works (Gao et al., 2024; Hassan et al., 2024) mainly adopt Frechet Video Distance (FVD) and Average Displacement Error (ADE) for trajectory alignment, while GEM (Hassan et al., 2024) adds human video evaluations that are subjective and hard to scale. The closest effort, ACT-Bench (Arai et al., 2024), focuses solely on trajectory alignment and overlooks key aspects such as video and trajectory distribution, quality, and temporal consistency.

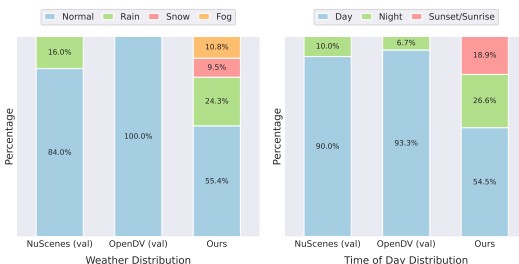

(a) Weather and time of day distribution in our ego-condition track.

(b) Data Ratio from existing open-sourced driving dataset in our ego-condition track.

| Dataset Source | Region | Ratio |
|---|---|---|
| Zod | Europe | 26.5% |
| Drivingdojo | China | 25.6% |
| CoVLA | Japan | 27.4% |
| Nuplan | U.S. | 8.8% |
| WOMD | U.S. | 11.6% |

Figure 3: The statistics of our ego-condition track.

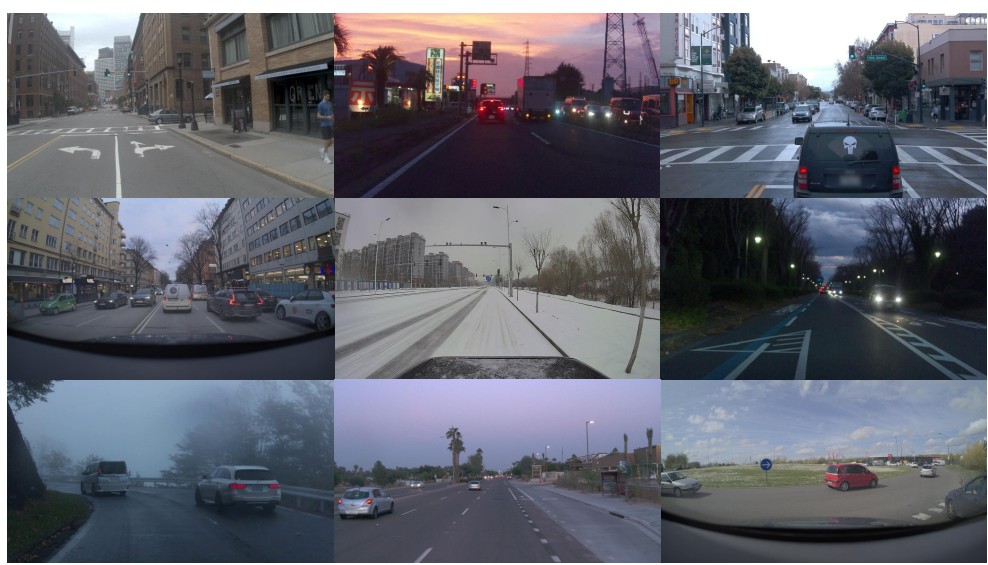

Figure 4: The gallery of our ego-condition track.

## B  APPENDIX

### B.1  GALLERY OF THE EGO-CONDITIONED TRACK

We present the distribution and gallery of our ego-conditioned track in Fig. 3 and Fig. 4. We curated data from five open-sourced driving datasets to diversify the distribution of weather, time of day, and locations (with various driving styles). The videos and ego-trajectories provided in these datasets are used as the target distribution for calculating metrics such as FVD and FTD.

### B.2  DETAILS OF OUR SLAM PIPELINE AND COMPARIISION WITH OTHERS

**Dealing with Unsuccessful Trajectory Reconstruction.** Not every generated video will yield a successful SLAM reconstruction, especially if the video has tremendous artifacts or very low texture. Simply discarding those cases would bias the evaluation, because typically it's the worst videos (the most unrealistic ones) that cause SLAM to fail. Dropping them would artificially inflate those poor-performing models' scores. We tackled this issue explicitly to ensure no video is left unevaluated. Our approach was to build a custom SLAM+depth estimation pipeline that is robust to failures. We ensure a trajectory is obtained for every video by applying a failure-recovery strategy: if at any frame the SLAM algorithm cannot estimate the next camera pose (*e.g.*, fails in feature matching, solving PnP, etc.), we take the last known pose and extrapolate it forward. Specifically, we propagate the last pose with a constant velocity model. To avoid giving an unrealistic advantage, we add

| Pipeline | Success rate ↑ | ADE ↓ |
|---|---|---|
| GEM: DROID-SLAM + Depth-Anything v2 | 17 / 20 | 14.61 |
| DrivinDojo: COLMAP + scale to GT | 16 / 20 | 14.99 |
| Ours w/o failure handling | 17 / 20 | 15.18 |
| Ours w/ failure handling | 20 / 20 | 16.84 |

Table 4: Comparison of different SLAM pipelines on 20 nuPlan videos generated with Vista. "Success rate" counts how many videos yield a valid reconstruction; ADE is the mean trajectory error over successfully reconstructed runs.

small random perturbations to the pose orientation during this extrapolation. This injects a bit of uncertainty to mimic the fact that the current estimation is noisy, preventing the extrapolated path from appearing "too perfect" in our metrics. We chose not to simply freeze the camera (no movement) upon failure, because a completely static continuation could skew certain trajectory metrics. By using this continuous-and-jitter method, we obtain a complete trajectory from start to end for every video, no matter how poor its quality. This allows all videos to count toward the trajectory-based metrics, holding models accountable for cases where a naive SLAM would have given up.

**Comparison with Other SLAM Pipelines.** We evaluated our reconstruction pipeline against those used in recent driving world-model systems. Concretely, we compare the successful reconstruction rate and trajectory accuracy (ADE) on 20 nuPlan videos generated with Vista from our early experiments. A run is counted as successful if the SLAM system returns a valid camera trajectory without numerical failure. The results are summarized in Table 4. Compared to the GEM pipeline (DROID-SLAM (Teed & Deng, 2022) + Depth-Anything v2 (Yang et al., 2024c)) and the DrivinDojo pipeline (COLMAP (Schönberger et al., 2016; Schönberger & Frahm, 2016) with scale aligned to ground truth), our basic version (*Ours w/o failure handling*) achieves a similar successful reconstruction rate (17/20 vs. 17/20 and 16/20) and a comparable ADE (15.18 vs. 14.61 and 14.99). When we enable our failure-handling strategy (*Ours w/ failure handling*), the successful rate increases to 20/20, while the ADE remains in the same ballpark (16.84). This trade-off is important for DrivingGen: the benchmark needs robust reconstruction on all videos rather than dropping harder cases and evaluating on a subset of "easy" videos. Overall, our SLAM pipeline is more robust than existing pipelines by handling reconstruction failure explicitly.

### B.3 FRÉCHET TRAJECTORY DISTANCE (FTD)

**Idea.** FTD applies the FID-style Gaussian Fréchet distance to *trajectory* embeddings, replacing image/video features with a driving-domain encoder.

**Representation model and input.** We use MTR's `agent_polyline_encoder` $\phi(\cdot)$ (Shi et al., 2023). *Crucially, MTR consumes a fixed temporal horizon $H$.*

**Window embeddings & trajectory pooling.** We slice the trajectory into windows to fit into the MTR encoder. Each window is encoded as $\mathbf{f} = \phi(\text{window}) \in \mathbb{R}^d$. A trajectory's embedding is the mean over its window embeddings, which stabilizes statistics and removes dependence on the number of windows.

**Distributional distance.** For generated embeddings $X = \{\bar{\mathbf{f}}(\tau_i^{\text{gen}})\}_{i=1}^n$ and reference embeddings $Y = \{\bar{\mathbf{f}}(\tau_j^{\text{ref}})\}_{j=1}^m$ with empirical means/covariances $\hat{\boldsymbol{\mu}}_{X/Y}, \hat{\Sigma}_{X/Y}$, define

$$\text{FTD}(X, Y) = \|\hat{\boldsymbol{\mu}}_X - \hat{\boldsymbol{\mu}}_Y\|_2^2 + \text{Tr}\left(\hat{\Sigma}_X + \hat{\Sigma}_Y - 2\big(\hat{\Sigma}_X^{1/2}\hat{\Sigma}_Y\hat{\Sigma}_X^{1/2}\big)^{1/2}\right)$$

We add $\varepsilon I$ ($\varepsilon=10^{-6}$) before the matrix square root and symmetrize products by $(A+A^\top)/2$ if needed. Optional Ledoit–Wolf shrinkage can be used when $n$ or $m < d$.

**Practical recipe (defaults).**

- **Encoder:** MTR `agent_polyline_encoder`.
- **Horizon & slicing:** $H=10$ steps; stride $s=H$ (non-overlapping); same slicing for generated and reference.

- **Normalization:** agent-centric translation/rotation per window; MTR schema constants $(\ell, w, h) = (4.5, 2.0, 1.8)$ m; type=vehicle; validity=1.

- **Aggregation:** mean over a trajectory's window embeddings; FTD on the two sets of trajectory-level embeddings.

### B.4 OBJECTIVE IMAGE QUALITY

**Motivation and background.** Pulse–width modulation (PWM) in vehicle lighting and roadside luminaires induces temporal luminance modulation that, when sampled by rolling-shutter cameras, can alias into low-frequency flicker and degrade detection and tracking. The IEEE Automotive P2020 standard formalizes *Modulation Mitigation Probability (MMP)* to quantify whether such modulation is sufficiently suppressed during operation (Group et al., 2018; 996, 2022). We implement MMP on the frame-mean luminance to provide a robust and efficient evaluation signal.

**Definition.** Given frames $\{I_t\}_{t=1}^T$ at sampling rate fps, form the luminance sequence $L_t = \text{mean}(\text{gray}(I_t))$ and its periodogram $\widehat{P}(f) = |\mathcal{F}\{L\}(f)|^2$ (real FFT). Let the dominant non-DC peak be

$$f^\star = \arg\max_{f>0} \widehat{P}(f).$$

If $f^\star < 0.2\,\text{Hz}$, set MMP $= 1$.

**Computation.** With the band $B(f^\star) = \{f : |f - f^\star| < \Delta f\}$, define the band-power ratio

$$A = \frac{\sum_{f \in B(f^\star)} \widehat{P}(f)}{\sum_f \widehat{P}(f) + \varepsilon}, \qquad \varepsilon = 10^{-8}.$$

The metric is

$$\boxed{\text{MMP} = \mathbf{1}[A < \tau]} \in \{0, 1\}.$$

**Defaults.** $\Delta f = \texttt{band\_hz} = 0.5\,\text{Hz}$, $\tau = \texttt{thr} = 0.05$, fps $= 10$. The procedure uses a single FFT per clip with complexity $O(T \log T)$.

### B.5 TRAJECTORY QUALITY

**Motivation.** Video-only scores can miss whether *motions* are plausible and comfortable. We define a trajectory quality that aggregates three kinematic submetrics—comfort, motion, and curvature—via a weighted geometric mean (equal weights by default). Each submetric lies in $[0, 1]$ with larger being better; we report per-trajectory scores and dataset means, skipping NaNs.

**Preliminaries.** A trajectory $\tau = \{(x_t, y_t)\}_{t=1}^T$. Velocities, accelerations, and jerks use centered finite differences. Heading comes from velocity, and yaw rate uses wrapped heading differences. Path length is the cumulative step distance. A trajectory is marked *moving* if any speed exceeds $v_{\text{static}} = 0.1$ m/s.

**Comfort** ($S_{\text{comf}}$). We score comfort from three per-meter peaks: longitudinal jerk, lateral acceleration, and yaw rate. Trajectories that are non-moving (speed $< v_{\text{static}}$) or too short ($\leq 1\,\text{m}$) are set to NaN. Each peak is then mapped to a $[0, 1]$ component score with an inverse transform $S_q = 1/(1 + q/s_q)$ (higher is better), where $s_q$ are scale factors (default 1.0). The final comfort score is the geometric mean of the three components.

**Motion** ($S_{\text{speed}}$). We penalize under-mobility using a trajectory's mean speed. A monotone log mapping compresses high speeds and scales by $v_{\text{max}} = k\, v_{\text{ref}}$ (defaults: $v_{\text{ref}} = 6.0$ m/s, $k = 2.5$) to obtain $S_{\text{speed}} \in [0, 1]$. Never-moving trajectories receive 0.

**Curvature** ($S_{\text{curv}}$). Discrete curvature is formed from first/second derivatives of $(x_t, y_t)$. We then compute an RMS curvature $\kappa_{\text{rms}}$, then map

$$S_{\text{curv}} = \frac{1}{1 + \kappa_{\text{rms}}} \in (0, 1].$$

Non-moving trajectories return NaN.

| Component | Example | Approx. Time |
|---|---|---|
| Video Generation | Wan2.2-14B
Vista | Days
About One Day |
| Trajectory Reconstruction | SLAM + Depth model | Hours |
| Distribution Metrics | FVD
FTD | Hours
Minutes |
| Quality Metrics | Subjective Image Quality
Objective Image Quality
Trajectory Quality | Hours
Minutes
Minutes |
| Consistency Metrics | Video Consistency
Agent Consistency
Agent Disappearance Consistency
Trajectory Consistency | Hours
More Hours
Hours
Minutes |
| Trajectory Alignment Metrics | ADE
DTW | Minutes
Minutes |
| All Metrics (Total) | All Above Metric Groups | 1–2 Days on a Single GPU |

Table 5: Approximate runtime of different components in DrivingGen on 400 videos with 100 frames each, evaluated on a single modern GPU. Times are coarse estimates and may vary with hardware.

### B.6 AGENT ABNORMAL DISAPPEARANCE

**Motivation.** Agents should not vanish without a plausible cause (e.g., occlusion or leaving the view). We detect such cases directly from video with a minimal vision–language check.

**Method.** For each agent that disappears, we prepare *three* frames: (1) the first frame where the agent is visible, (2) the last frame where it is visible (both with the agent box drawn in green), and (3) the first frame after it disappears (no box). We ask a VLM to classify the disappearance with the following prompt:

```
Given three frames around the moment a green-boxed object
disappears, classify the disappearance as Natural (e.g.,
occlusion or leaving the field of view) or Unnatural (abrupt
or non-physical).  Base your decision on visual and motion
continuity and interactions with nearby objects.  Output one
word:  Natural or Unnatural.
```

**Scoring.** A tracklet is *abnormal* if the VLM outputs `Unnatural`; otherwise it is *not abnormal*. A video is *clean* only if all evaluated tracklets are *not abnormal*. The final score is the percentage of clean videos (higher is better).

### B.7 TRAJECTORY CONSISTENCY

**Definition.** From positions sampled at step $\Delta t$, form the speed series $v_t$ and the acceleration series $a_t$ by finite differences. Measure each signal's dispersion relative to its typical level using a simple ratio, then squash with an exponential:

$$R_v = \frac{\text{std}(v)}{\text{mean}(v)}, \quad R_a = \frac{\text{std}(a)}{\text{mean}(|a|)}, \qquad S_v = \exp(-R_v), \ S_a = \exp(-R_a).$$

The trajectory consistency score is the average

$$S_{\text{cons}} = \tfrac{1}{2}\left(S_v + S_a\right) \in (0, 1],$$

where higher indicates smoother, more realistic kinematics.

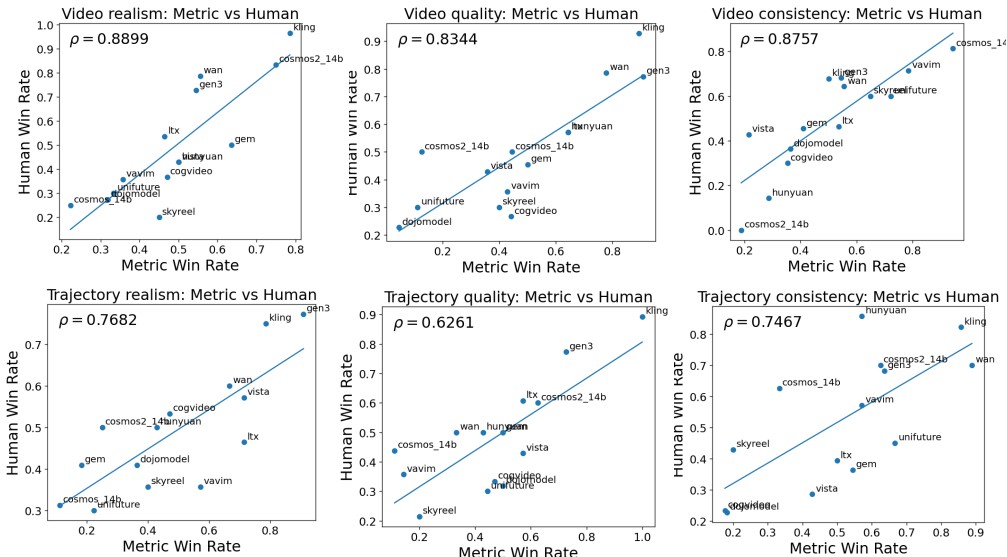

Figure 5: Human Validation of Our benchmark. Our metrics closely match human preferences. Trajectory-related metrics are less accurate in comparison to humans, likely due to noisy monocular SLAM and metric-depth recovery from generated videos with artifacts.

## B.8 TIME AND RESOURCE FOR DRIVINGGEN

In our experiments, the bottleneck is primarily the video generation itself: many of the state-of-the-art generative models we benchmark are slow and memory hungry (e.g., Wan2.2-14B takes about 20-30 minutes to generate one 100-frame video on a single GPU with at least 40 GB memory). In contrast, the evaluation suite is comparatively manageable. The approximate wall-clock time for each metric group on 400 videos is summarized in Table 5. On a single modern GPU, running all metrics for 400 videos with 100 frames takes roughly 1–2 days.

Within this budget, the main cost on the evaluation side comes from image quality and video consistency metrics, which require running heavy visual backbones over every frame. The most time-consuming metrics would be agent consistency and disappearance consistency, which run models for each agent in the first frame of the video. Trajectory measures (FTD, quality, consistency and alignment) are much cheaper (minutes), since they operate on compact embeddings or low-dimensional trajectories. These numbers are indicative and may vary with hardware and implementation, but they show that: (i) video generation dominates the overall runtime, and (ii) among the metrics, the image, video and agent quality and consistency components are the main contributors, while the rest of the metrics are comparatively fast.

## B.9 HUMAN ALIGNMENT OF DRIVINGGEN

We employ a similar method in VBench to determine whether each category aligns with human preferences. Given the human labels, we calculate the win ratio of each model. During pairwise comparisons, if a model's video is selected as better, then the model scores 1 and the other model scores 0. If there is a tie, then both models score 0.5. For each model, the win ratio is calculated as the total score divided by the total number of pairwise comparisons in which it participated.

For fast and reasonable evaluation, we select three categories: distribution, quality and consistency. We evaluate with both videos and trajectories and use the primary metric in each category. Metrics are FVD and FTD, Subjective image quality and trajectory quality, video consistency and trajectory consistency. The results are shown in Fig. 5.

