# OpenReview forum: "DrivingGen: A Comprehensive Benchmark for Generative Video World Models in Autonomous Driving"
_ICLR.cc/2026/Conference — ICLR 2026 Poster_

### Official Review · Reviewer_7qoX · 2025-10-22

**Soundness:** 3
**Presentation:** 3
**Contribution:** 3
**Rating:** 8
**Confidence:** 5

**Summary:**

This work focuses on advancing world models for autonomous driving by introducing a new benchmark, DrivingGen. Unlike previous evaluations that primarily assess visual quality, DrivingGen offers a fine-grained assessment of physical realism, safety, and trajectory consistency, establishing a unified evaluation framework to promote the development of reliable, controllable, and deployable driving world models. Through benchmarking 14 state-of-the-art models, our study reveals clear trade-offs: general-purpose models produce visually appealing results but often violate physical constraints, whereas driving-specific models capture realistic motion dynamics yet fall short in visual quality.

**Strengths:**

1. The motivation is clear and valuable for advancing future driving world models.
2. The authors present a thorough empirical study: they evaluate a wide range of methods.
3. Comprehensive evaluation framework: by assessing methods along multiple dimensions — distribution, quality, temporal alignment, and trajectory alignment.

**Weaknesses:**

1. Although the metrics are intended for evaluating world models, most of them still focus on visual quality. There are relatively few metrics that assess whether the state transformations of the world model are reasonable and accurate.
2. Counterfactual reasoning is also an important capability of a world model. It could be interesting to consider evaluating this aspect in future work.
3. The evaluation of physical reasonable is somewhat lacking. It might be useful to draw inspiration from VBench 2.0 and incorporate metrics.

**Questions:**

1. How to deal with cases where the generated video fails to reconstruct trajectories? SLAM-based methods, do not successfully produce a trajectory for every video sequence.
2. DrivingDojo proposes the AIF metric to measure trajectory alignment. It might be worth discussing the advantages and limitations of the trajectory metrics proposed in this paper.
3. What is the approximate time and resource cost for evaluating 400 videos? This is also an important consideration when assessing the practicality of a benchmark.
4. AVG Ranking may not be an appropriate metric. Could a composite, weighted total score be designed to more comprehensively evaluate world model performance

---

> ### Author Response · Authors · 2025-11-20
> **Response to Reviewer 7qoX (1/5)**
>
> Dear reviewer `7qoX`, we sincerely thanks for your thorough assessment and contributive suggestions on our paper! We address each of your questions as follows.
>
> > Although the metrics are intended for evaluating world models, most of them still focus on visual quality. There are relatively few metrics that assess whether the state transformations of the world model are reasonable and accurate.
>
> Thanks for pointing out this. For assessing whether the state transformations of the world model are reasonable and accurate, we focus on whether the *temporal evolution* of the ego-trajectory and other agents remains physically plausible. In DrivingGen, trajectory quality metrics (smoothness, curvature, and kinematic feasibility) constrain the ego vehicle’s accelerations and turning rates, while our agent-level consistency and abnormal disappearance metrics penalize objects that abruptly change speed, or vanish without cause.
>
> Designing richer tests of state transformations that go beyond kinematics (e.g., fine-grained scene dynamics or rule-based traffic behavior) would require controllable generative models and datasets with detailed state annotations. Since most current driving world models do not yet support such scene controls (see our response to Reviewer `XepP`’s question of scene controlability), we leave these more advanced physics-based evaluations as important future work, which we now discuss in the “scene controllability and state transformations” paragraph of our new “future work and limitations” section (marked as blue).
>
> > Counterfactual reasoning is also an important capability of a world model. It could be interesting to consider evaluating this aspect in future work.
>
> This is a great question! We agree that counterfactual reasoning is an intriguing capability for a world model – e.g., being able to imagine “what if” scenarios or alter certain events in the video to see different outcomes. In our current benchmark, we did not explicitly evaluate counterfactual reasoning. The main reason is that DrivingGen focuses on real driving videos and ground-truth trajectories, so we are limited to evaluating the scenarios that actually happened. To test counterfactuals, one might need to introduce hypothetical events or modifications (e.g., ask the model to imagine if a pedestrian had stepped out, or if the ego vehicle had taken a different turn). Implementing this uniformly would be quite complex and was beyond our scope. Additionally, most driving video generation models today do not support direct counterfactual inputs as noted above. For instance, we do not have driving world models that take a text like “now imagine the car runs a red light” as input. Thus, we considered counterfactual reasoning out of scope for the initial benchmark.
>
> We do find this idea very interesting and agree it could reveal important capabilities and have added it to the new section “future work and limitations” in the revised manuscript (marked as blue). In future work, we may incorporate a counterfactual evaluation component, perhaps by introducing controlled variations of scenarios (e.g., via simulation or editing real videos) to see if models can generate the altered outcome. This would likely require new data or simulator support. One can introduce hypothetical events or modifications (like an astronaut on a horse crossing the road, or a car jumping off the ground to overtake other agents, and other unrealistic edge cases) and propose new metrics to check whether the model follows this counterfactual generation.

---

> ### Author Response · Authors · 2025-11-20
> **Response to Reviewer 7qoX (2/5)**
>
> > The evaluation of physical reasonable is somewhat lacking. It might be useful to draw inspiration from VBench 2.0 and incorporate metrics.
>
> Thanks for the suggestion. As mentioned above, we do incorporate several physical realism checks in DrivingGen, though we may not have framed them exactly as in VBench2.0. Concretely, we evaluate the smoothness and feasibility of the ego-vehicle trajectory (motion plausibility, kinematic constraints) and penalize discontinuities in agent motion (agent stability, agent disappearance). In VBench 2.0, the physical metrics include categories state transformation (mechanics, thermotics, material) and view consistency. Part of those (e.g., thermotics, material) are not directly applicable to driving videos and may need further exploration. However, the mechanical consistency aspects are relevant, and indeed our metrics address them in trajectories. For multi-view consistency, object stability and motion speed consistency are measured as similarly to the implementation in VBench 2.0: if a generated car’s motion is erratic, the trajectory smoothness metric and our agent velocity statistics would reflect that. If an agent ghosts out, our disappearance metric catches that, akin to testing object permanence.
>
> In summary, while DrivingGen might appear to emphasize visual quality, it does evaluate several physical aspects of the world model’s outputs. We believe our current evaluation of trajectory quality, consistency, and agent permanence addresses the core of physical reasonableness for driving scenes, but we agree that expanding this angle is beneficial as the benchmark evolves.

---

> ### Author Response · Authors · 2025-11-20
> **Response to Reviewer 7qoX (3/5)**
>
> > How to deal with cases where the generated video fails to reconstruct trajectories? SLAM-based methods, do not successfully produce a trajectory for every video sequence.
>
> Thanks for this careful review and it's a valuable question! Not every generated video will yield a successful SLAM reconstruction, especially if the video has tremendous artifacts or very low texture. We tackled this issue explicitly to ensure no video is left unevaluated. Our approach was to build a custom SLAM+depth estimation pipeline that is robust to failures.
>
> Simply discarding those cases would bias the evaluation, because typically it’s the *worst* videos (the most unrealistic ones) that cause SLAM to fail. Dropping them would artificially inflate those poor-performing models’ scores. Instead, we ensure a trajectory is obtained for every video by applying a failure-recovery strategy: if at any frame the SLAM algorithm cannot estimate the next camera pose, we take the last known pose and extrapolate it forward. In practice, we propagate the last pose with constant velocity. To avoid giving an unrealistic advantage (a perfectly smooth straight-line trajectory), we add small random perturbations to the pose orientation during this extrapolation. This injects a bit of uncertainty to mimic the fact that the current estimation is noisy, preventing the extrapolated path from appearing “too perfect” in our metrics. We chose not to simply freeze the camera (no movement) upon failure, because a completely static continuation could skew certain trajectory metrics. By using this continuous-and-jitter method, we obtain a complete trajectory from start to end for every video, no matter how poor its quality. This allows all videos to count toward the trajectory-based metrics, holding models accountable for cases where a naive SLAM would have given up.
>
> We did an early experiment with SLAM pipelines. DrivingDojo (COLMAP + scale to ground truth) and GEM (Droid slam + Depth Anything v2) are considered. As presented below, our basic version (Ours w/o failure handling) achieves a similar successful reconstruction rate (17/20 vs.\ 17/20 and 16/20) and a comparable ADE (15.18 vs.\ 14.61 and 14.99). When we enable our failure-handling strategy (Ours w/ failure handling), the success rate increases to 20/20, while the ADE remains in the same ballpark (16.84). This trade-off is important for DrivingGen: the benchmark needs robust reconstruction on all videos rather than dropping harder cases and evaluating on a subset of “easy’’ videos.
>
> | Pipeline                            | Success rate ↑ | ADE ↓ |
> | ----------------------------------- | -------------- | ----- |
> | GEM: DROID-SLAM + Depth-Anything v2 | 17 / 20        | 14.61 |
> | DrivinDojo: COLMAP + scale to GT    | 16 / 20        | 14.99 |
> | Ours w/o failure handling           | 17 / 20        | 15.18 |
> | Ours w/ failure handling            | 20 / 20        | 16.84 |
>
> Overall, our SLAM pipeline is more robust than existing pipelines by handling reconstruction explicitly. We have clarified this procedure in the revised manuscript (in lines 302-304 and Appendix A.2). Thanks for the comments.
>
> > DrivingDojo proposes the AIF metric to measure trajectory alignment. It might be worth discussing the advantages and limitations of the trajectory metrics proposed in this paper.
>
> Thank you for the question. DrivingDojo introduced the action instruction following (AIF) metric to measure how well a generated trajectory aligns with the intended route. And the mean value of absolute error (L1 norm) is used, while we share a similar procedure but use the ADE (L2 norm).
>
> In addition to the strict frame-by-frame alignment error, we use *the Dynamic Time Warping (DTW)* metric for complementary evaluation. The DTW metric evaluates how well the generated path matches the ground-truth route shape, allowing for slight timing discrepancies. The motivation is that a good world model might follow the correct road/path but slightly lag or lead in time; DTW can forgive small temporal misalignments and focus on spatial alignment. This is a complementary perspective to strict frame-by-frame alignment. By combining these, we provide a more comprehensive trajectory fidelity assessment.
>
> The limitation of our approach (with other SLAM-based approaches mentioned above) is that it depends on the accuracy of the vision algorithms (e.g., SLAM, depth), biases or errors that will reflect in the metrics. However, since all models are evaluated with the same pipeline, any systematic bias is shared, and model-to-model comparisons remain fair. Overall, our trajectory alignment metrics serve a similar purpose to AIF, but are more comprehensive and suited to a unified benchmark setting.

---

> ### Author Response · Authors · 2025-11-20
> **Response to Reviewer 7qoX (4/5)**
>
> > What is the approximate time and resource cost for evaluating 400 videos? This is also an important consideration when assessing the practicality of a benchmark.
>
> This is an important question. As briefly mentioned in lines 262-265 of our manuscript, 400 is a number striking for a balance between efficiency and meaningful evaluation.
>
> In our experiments, the bottleneck is primarily the video generation itself: many of the state-of-the-art generative models we benchmark are slow and memory hungry (e.g., Wan2.2-14B takes about 20-30 minutes to generate one 100-frame video on a single GPU with at least 40 GB memory). In contrast, the evaluation suite is comparatively manageable. The approximate wall-clock time for each metric group on 400 videos is summarized in the following Table. On a single modern GPU, running all metrics for 400 videos with 100 frames takes roughly 1–2 days.
>
> | Component                    | Example                         | Approx. Time              |
> | ---------------------------- | ------------------------------- | ------------------------- |
> | Video Generation             | Wan2.2-14B                      | Days                      |
> |                              | Vista                           | About One Day             |
> | Trajectory Reconstruction    | SLAM + Depth model              | Hours                     |
> | Distribution Metrics         | FVD                             | Hours                     |
> |                              | FTD                             | Minutes                   |
> | Quality Metrics              | Subjective Image Quality        | Hours                     |
> |                              | Objective Image Quality         | Minutes                   |
> |                              | Trajectory Quality              | Minutes                   |
> | Consistency Metrics          | Video Consistency               | Hours                     |
> |                              | Agent Consistency               | More Hours                |
> |                              | Agent Disappearance Consistency | Hours                     |
> |                              | Trajectory Consistency          | Minutes                   |
> | Trajectory Alignment Metrics | ADE                             | Minutes                   |
> |                              | DTW                             | Minutes                   |
> | All Metrics (Total)          | All Above Metric Groups         | 1--2 Days on a Single GPU |
>
> Within this budget, the main cost on the evaluation side comes from image quality and video consistency metrics, which require running heavy visual backbones over every frame. The most time-consuming metrics would be agent consistency and disappearance consistency, which run a backbone model (e.g., DINO, Cosmos-Reason-1) for each agent in the first frame of the video. Distribution-level statistics, trajectory measures (quality, consistency and alignment) are much cheaper (in minutes), since they operate on compact embeddings or low-dimensional trajectories. These numbers are indicative and may vary with hardware and implementation, but they show that: 1) video generation often dominates the overall runtime, and 2) among the metrics, the image, video and agent quality and consistency components are the main contributors, while the rest of the metrics are comparatively fast.
>
> We have added the summary and the table of the time and resource requirements in the revised manuscript for clarity (see lines 411-412 and Appendix A.8). The key point is that DrivingGen is meant to be reproducible and not prohibitively expensive. Researchers can run the evaluation on their own model outputs within, say, 1-2 days on a single machine. We believe this cost is justified by the richness of the metrics.

---

> ### Author Response · Authors · 2025-11-20
> **Response to Reviewer 7qoX (5/5)**
>
> > AVG Ranking may not be an appropriate metric. Could a composite, weighted total score be designed to more comprehensively evaluate world model performance.
>
> Thank you for highlighting this concern. In the manuscript, we reported an average rank mainly to give a coarse overall picture, while still presenting the detailed breakdown on each metric. It’s not an official metric in our evaluation pipeline. We appreciate the reviewer’s perspective on this and have emphasized in the revised text that the overall ranking should be interpreted with caution, whereas the disaggregated metrics give a fuller picture (see lines 416-417).
>
> Designing a composite “overall score” is appealing but comes with challenges: different metrics have different scales and significance, thus more analysis is needed for a fair comparison. For quality and temporal consistency metrics, the results are scaled from 0 to 1, but the absolute value is not necessarily comparable (e.g., subjective image quality mainly ranges from 0.4 to 0.6, while video consistency normally ranges above 0.8). For distribution and trajectory alignment metrics, the situation is more complicated. They calculate on different scales and ranges, which makes it hard to normalize and compare. For example, Frechet Video Distance (FVD) is unbounded and not linearly comparable to, say, trajectory ADE, so combining them requires careful normalization and weighting. Any weighting scheme would introduce subjective bias, essentially deciding that “X points of FVD = Y meters of ADE”, which is hard to justify objectively.
>
> Moreover, we observed that in our results, models that do well in one category may trade off another (e.g., some models have excellent visual quality but poor physical metrics). A single composite score might obscure these trade-offs. Notably, some prior work in video generation benchmarks do not report a single aggregate number for similar reasons (e.g., VBench 2.0). We are certainly open to more sophisticated aggregation methods suitable for driving related metrics in the future. For now, as the first benchmark of its kind, we believe providing the full table of metrics is the most transparent way to evaluate the models, and the average rank serves as a quick summary but not a definitive score.
>
> We consider the exploration of a composite, single-index performance metric as an open topic and have added it in the new section “future work and limitations” in our revised manuscript (marked as blue), one that could be discussed in a broader community setting to avoid ad-hoc solutions. For this work, sticking to the individual metrics seemed the most straightforward and unbiased approach.

---

> > ### Comment · Reviewer_7qoX · 2025-11-28
> >
> > I appreciate the authors' detailed response. While evaluation costs are considerable, it is the right direction for world model assessment. I maintain my score as Accept.

---

### Official Review · Reviewer_XepP · 2025-10-22

**Soundness:** 3
**Presentation:** 4
**Contribution:** 3
**Rating:** 6
**Confidence:** 5

**Summary:**

This paper focuses on evaluating driving video generation methods. Existing benchmarks emphasize image quality metrics (e.g., FID, FVD) and often fail to assess aspects critical to autonomous driving, such as the controllability of the ego vehicle's trajectory, the reasonableness of agent trajectories, and the consistency of 3D contents. Current benchmarks also lack data diversity in terms of weather, lighting, road conditions, and geographic locations. This work samples data from multiple datasets to enhance data diversity. proposes a set of metrics for evaluating video output, and benchmarks the performance of several state-of-the-art models.

**Strengths:**

Establishing a more practical benchmark for driving video generation is valuable. The contributions of providing more diverse test data and comprehensive evaluation metrics are clear. To maximize its contribution, the benchmark and evaluation code should be made open-source.

**Weaknesses:**

No major flaws were identified in the current work. However, the authors could further improve the benchmark by including evaluations for scene content controllability. While the paper addresses video quality, temporal consistency, and ego trajectory controllability, the controllability of generated scene contents( such as agents controlled with bounding boxes or roads via maps/lane) is also important for autonomous driving applications. These control signals can be extracted from videos using detection and segmentation methods if not available in the original datasets.

Typo: 'Qulity' in Table 2.

**Questions:**

see weakness

---

> ### Author Response · Authors · 2025-11-20
> **Response to Reviewer XepP**
>
> Dear reviewer `XepP`, We thank you for the thoughtful review and precious feedback on our paper! We have carefully addressed your concerns as outlined below.
>
> > No major flaws were identified in the current work. However, the authors could further improve the benchmark by including evaluations for scene content controllability. While the paper addresses video quality, temporal consistency, and ego trajectory controllability, the controllability of generated scene contents( such as agents controlled with bounding boxes or roads via maps/lane) is also important for autonomous driving applications. These control signals can be extracted from videos using detection and segmentation methods if not available in the original datasets.
>
> Thanks for sharing this interesting idea. We agree that evaluating controllability over scene content (e.g., controlling other agents, road layout, or specific objects in the scene) would be highly useful for autonomous driving applications. We did not include such metrics in our benchmark because implementing a unified evaluation for different models with scene-level control faces challenges: 1) *Model support:* Current generative models vary widely in whether and how they can take content-level control signals. Many models (e.g., general generative worlds) are not trained to follow explicit prompts for other agents (like “move this specific car to a location”) or map-based layouts. And four out of five considered driving world models don’t support these controls explicitly (only GEM and GAIA-2 (closed-source) support these controls and evaluations, as presented in Table 1 of our manuscript). We lacked a meaningful interface to test these models on agents and map layout controllability. (ii) *Dataset complexity:* Measuring scene controllability would involve bounding boxes and semantic maps from each test video and feeding them into models. This is infeasible for our open-domain track which is collected from the web to test the model’s generalizability. Given these difficulties, DrivingGen focused on ego-trajectory control as a first try, a form of controllability that all models supported to do a fair and unified evaluation (e.g., general models are trained with camera control and driving models are trained with ego-trajectory control).
>
> It is a great topic for structural driving world generation which controls scene content and map layout. We have mentioned the idea in the new section “future work and limitation” in our revised manuscript (marked as blue). Scene controllability evaluation requires harder constraints on models and datasets. One could imagine controlling the presence or behavior of a pedestrian or the configuration of lanes, and checking if the model can follow those constraints. This would require strong priors and structural inputs that not all current models have, but it’s an exciting direction. As more models incorporate such control explicitly, it’s critical to include content-level control evaluations in an expanded benchmark.
>
> > Typo: 'Qulity' in Table 2.
>
> Thank you for pointing this out. We have fixed "Quality" in Table 2 in our revised manuscript.

---

### Official Review · Reviewer_cB5B · 2025-10-31

**Soundness:** 3
**Presentation:** 3
**Contribution:** 3
**Rating:** 6
**Confidence:** 4

**Summary:**

The DrivingGen benchmark is introduced as the first comprehensive evaluation framework for generative world models in autonomous driving, targeting key gaps where existing methods overlook safety-critical factors, trajectory plausibility, temporal consistency, and motion controllability. DrivingGen utilizes a highly diverse dataset spanning varied weather, time of day, global regions, and complex maneuvers essential for robust deployment. It introduces a suite of novel, specialized metrics that jointly assess performance across four dimensions: visual realism, trajectory plausibility, temporal coherence, and control fidelity. Benchmarking 14 state-of-the-art models revealed a key finding: a trade-off exists where general models may appear visually superior but "break physics," while driving-specific models prioritize realistic motion accuracy but often lag in visual quality, thus establishing a unified framework to guide future development.

**Strengths:**

1. The dataset distributions are balanced and diverse across all conditions.

2. DrivingGen has created a new, specialized set of various measurements to evaluate generated driving videos; these are designed for the complexities of driving and are therefore more effective than standard video evaluation tools.

3. The experiments are comprehensive.

**Weaknesses:**

1. It would be more convincing to incorporate downstream task's performance (detection, mapping, planning) into the evaluation system. But this seems difficult since the data contains only front-view data.

**Questions:**

See weaknesses.

---

> ### Author Response · Authors · 2025-11-20
> **Response to Reviewer cB5B**
>
> Dear reviewer `cB5B`, we sincerely thank the careful review and valuable feedback on our paper! We have addressed your concerns as follows.
>
> > It would be more convincing to incorporate the downstream task's performance (detection, mapping, planning) into the evaluation system. But this seems difficult since the data contains only front-view data.
>
> Thanks for the insightful question. We appreciate the suggestion to incorporate downstream task metrics into the benchmark. Our primary goal in DrivingGen is to provide the first systematic benchmark that directly measures video realism, physical consistency, and controllability in the generated footage itself, with different generative world models to be considered. As presented in Table 1 of our manuscript, existing works that report downstream performance are all method works: they consider a single model so the extra training computational cost is manageable. In contrast, a benchmark work must treat many generative models fairly, which would multiply the computation far beyond what is typical for benchmark papers and is not done in prior video-generation benchmarks either. With 14 models to be considered in our benchmark, the computation and resources would be hugely increased. Moreover, as the Reviewer noted, our current dataset is limited to a single front-view camera feed, which poses challenges for such training and evaluations.
>
> We also want to share the motivation for a single front-view format. We aim to maximize benchmark generalization and inclusivity: it allows testing a wide range of generative world models under a common setting with diverse and meaningful data. In fact, several cutting-edge driving world models rely on multi-view inputs or HD maps (e.g., DriveDreamer and MagicDrive series), which are hard to evaluate on a benchmark with different world models and datasets (e.g., most of them are evaluated solely on nuScenes). By considering front-view only generation methods (we believe this is a trend starting from Vista-based methods), we ensured that both general/physical generative video models (e.g., Wan, Cosmos, etc.) and driving generative models can be evaluated together, and we could leverage diverse data (from the web or aggregate from multiple datasets) for evaluation rather than validation on specific dataset dominated by normal driving scenarios.
>
> A possible future direction is expanding the benchmark to multi-view video and sensor data (LiDAR, HD Map, etc.) to construct a more structured driving world generation and novel metrics of downstream tasks can be proposed. However, it requires collecting synchronized multi-camera footage and Map knowledge for a fair and meaningful benchmark. At the same time, the world model’s input should also consider these conditions to generate better driving scenarios, which current video generation models may not support (see our response to Reviewer `XepP`’s question about scene controllability). In brief, many of the current driving world models lack support for scene controls.
>
> We have added this idea, along with the downstream tasks evaluation, to our new section “future work and limitations” in our revised manuscript (marked as blue). We believe it’s insightful for readers to understand DrivingGen and follow our work.

---

### Official Review · Reviewer_2jMw · 2025-11-01

**Soundness:** 3
**Presentation:** 3
**Contribution:** 3
**Rating:** 6
**Confidence:** 4

**Summary:**

This paper introduces DrivingGen, a comprehensive benchmark designed to evaluate generative world models for autonomous driving. The authors argue that existing benchmarks are limited in scope, failing to capture the full requirements of driving-specific simulation—such as visual realism, trajectory plausibility, temporal consistency, and controllability—while also lacking diversity in weather, geography, and driving maneuvers.

**Strengths:**

- DrivingGen is the first benchmark to jointly evaluate visual, kinematic, and interactive aspects of generative driving world models, addressing critical gaps in prior works.
- Introduces FTD for trajectory distribution, kinematic quality scores, and agent disappearance detection using VLMs—all tailored for driving safety and realism.
- The dataset includes under-represented conditions (e.g., night, snow, sandstorms) and global geographic variety, enabling more robust and realistic evaluation.
- The authors validate metric alignment with human preferences, increasing confidence in the benchmark’s practical relevance.

**Weaknesses:**

- With only 400 clips, the dataset may not fully represent the long-tail of real-world driving scenarios, despite its diversity.
- The benchmark focuses on open-loop video generation and does not assess models in interactive, closed-loop simulation settings.

**Questions:**

- Why was the dataset limited to 400 clips? Are there plans to expand it in the future to better cover rare and safety-critical scenarios?
- How do you mitigate the impact of video artifacts on SLAM and depth estimation, especially for models with low visual fidelity?

---

> ### Author Response · Authors · 2025-11-20
> **Response to Reviewer 2jMw (1/2)**
>
> Dear reviewer `2jMv`, we sincerely appreciate the detailed assessment and valuable feedback on our paper! Here, we provide responses and explanations to your comments and suggestions.
>
> > With only 400 clips, the dataset may not fully represent the long-tail of real-world driving scenarios, despite its diversity.
> >
> > and
> >
> > Why was the dataset limited to 400 clips? Are there plans to expand it in the future to better cover rare and safety-critical scenarios?
>
> Thanks for the question. We acknowledge the concern that 400 clips may not fully cover the long tail of driving scenarios. Although 400, we still try our best to control the percentage of normal scenarios and cover many rare and safety-critical real-world driving scenarios (e.g., varied weather, time of day, geographic, and complex events) as mentioned in our manuscript.
>
> We chose 400 as a balance between efficiency and practicality (as briefly mentioned in lines 262-265 in our manuscript), because generating and evaluating videos is resource-intensive (see our response to Reviewer `7qoX`’s question about the practicality of our benchmark), so a moderate number ensures efficient testing and iteration and is accessible for academic researchers without extraordinary resources to use and expand. In our experience, 400 provided enough samples to build a data distribution we find reasonable and distinguish from previous datasets for validating the generative driving model’s ability.
>
> At the same time, there are several existing video benchmarks (e.g., VBench, WorldModelBench) that also use on the order of a few hundred samples, driven by similar considerations for academic practical use. As DrivingGen is the first benchmark of its kind, we believe 400 clips provide meaningful evaluation coverage while remaining accessible. In future expansions, we plan to scale up the dataset to include even more long-tail cases as models and evaluation resources improve. As generative models become faster and datasets become more readily available, scaling up to thousands of clips is feasible and will further improve long-tail coverage. This initial 400 is a foundation on which we and others can build in follow-up work.
>
> We’ve added the idea of expanding more meaningful data in the new section “future work and limitations” in our revised manuscript (marked as blue). We believe it’s insightful for readers to understand DrivingGen and follow our work.
>
> > The benchmark focuses on open-loop video generation and does not assess models in interactive, closed-loop simulation settings.
>
> Thanks for this constructive suggestion. We agree that interactive, closed-loop simulation is an important goal for driving world models. In the current work, all considered generative video world models are designed for open-loop video generation (sequence prediction given initial conditions) because truly closed-loop generation and simulation for world models pose significant challenges at this stage (see our response to Reviewer `XepP`’s question about scene controllability). In brief, many of the current driving world models lack support for scene controls. Existing generative models have diverse architectures and inputs, and no standardized closed-loop world generation framework exists yet; thus performing a fair, unified closed-loop benchmark would be nontrivial.
>
> Moreover, our open-loop setting still involves multi-round predictions: models generate sequences of 100 frames by recursively feeding their outputs at the current round as the input of the next round (e.g., Vista and GEM generate 25 frames per round and the last 3 frames are used to generate the next 25 frames), which simulates some feedback loop behavior. We consider fully closed-loop evaluation (e.g., integrating generative models into an interactive simulator like CARLA) to be beyond our current scope due to practical limitations.
>
> We absolutely agree that it is an exciting future direction. Ensuring reliable closed-loop performance (e.g., for safe planning) is indeed crucial, and DrivingGen is a step toward that by first benchmarking open-loop predictive quality and realism. We have added the idea of closed-loop evaluation for driving world models in the new section “future work and limitations” in our revised manuscript (marked as blue).

---

> ### Author Response · Authors · 2025-11-20
> **Response to Reviewer 2jMw (2/2)**
>
> > How do you mitigate the impact of video artifacts on SLAM and depth estimation, especially for models with low visual fidelity?
>
> Thanks for the careful review! In our evaluation pipeline, all models’ videos go through the same SLAM+depth pipeline, so if a model produces severe visual artifacts (e.g., flicker, distortion, noise blocks), the pipeline might struggle to reconstruct good trajectories. However, this equally affects all models and thus reflects the model’s performance. We do not filter out or manually correct trajectories for videos with artifacts; instead, a model that produces poor visual fidelity will naturally get worse trajectory metrics, which we view as a fair outcome (poor video quality leading to poor reconstructed trajectory indicates the model breaks the physical realism).
>
> However, to ensure every video yields a trajectory (e.g., when the video is extremely artifact-laden), we implemented a robust fallback in our SLAM pipeline (explained in detail in our response to Reviewer `7qoX`’s question about dealing with trajectory reconstruction). In brief, if the SLAM pipeline fails at some frame, we propagate the last known camera pose forward with slight random perturbations. This guarantees a complete trajectory for each video while avoiding overly smooth trajectories to hack metrics. Apart from this, we do not apply special mitigation for low-fidelity videos. Importantly, since the SLAM and depth models are fixed and identical for all evaluations, any degradation in trajectory accuracy due to artifacts is a consistent penalty incurred by the models that produce such artifacts. This consistency ensures the comparison remains fair: video artifacts simply indicate a model’s limitations, and our benchmark fairly captures that in the results.

---

### Meta-Review · Area_Chair_aBG9 · 2025-12-29

**Summary:**

This work presents a good benchmark for assessing generative driving world (video generation) models, with focus on a number of of important aspects such as trajectory quality, temporal coherence, controllability. It contributes a dataset for evaluation (400 videos), along with new metrics. Totally 14 SOTA models are evaluated under this benchmark, setting a great ground for the future assessment and comparison of driving world models for the community.

**Reviewer Concerns:**

One main issue is about the setting of this benchmark which is only limited to open-loop assessment, rather than closed-loop setting which is more realistic. The authors agree with this and aim to extend it in the future when more advanced world models are available. Given the literature, this makes sense. The authors have discussed this point.

Other issues like why only 400 videos for test, evaluation cost, impact of video artefacts, similarity with other existing metrics, failure cases, have been well addressed in the revision.

**Reviewer Scores:**

Given these responses, all reviewers should at least maintain their scores.

---

### Decision · Program_Chairs · 2026-01-26

Accept (Poster)